# Long-Term Survival Following Minimally Invasive Lung Cancer Surgery: Comparing Robotic-Assisted and Video-Assisted Surgery

**DOI:** 10.3390/cancers14112611

**Published:** 2022-05-25

**Authors:** François Montagne, Zied Chaari, Benjamin Bottet, Matthieu Sarsam, Frankie Mbadinga, Jean Selim, Florian Guisier, André Gillibert, Jean-Marc Baste

**Affiliations:** 1Department of Thoracic Surgery, CHU Lille, F-59000 Lille, France; francois.montagne0438@orange.fr; 2Department of Thoracic and Cardiovascular Surgery, University of Sfax, Habib Bourguiba University Hospital, Sfax 3029, Tunisia; chaari.zied1@gmail.com; 3Department of General and Thoracic Surgery, Rouen University Hospital, 1 Rue de Germont, F-76000 Rouen, France; benjamin.bottet@chu-rouen.fr (B.B.); matthieu.sarsam@chu-rouen.fr (M.S.); frankie.mbadinga@chu-rouen.fr (F.M.); 4Department of Anesthesiology and Critical Care, CHU Rouen, F-76000 Rouen, France; jean.selim@chu-rouen.fr; 5Normandie University, University of Medicine and Pharmacy of Rouen, UNIROUEN, INSERM U1096, FHU REMOD-VHF, F-76183 Rouen, France; 6Thoracic Oncology and Respiratory Intensive Care Unit, Department of Pneumology, Rouen University Hospital, F-76000 Rouen, France; florian.guisier@chu-rouen.fr; 7Normandie University, University of Medicine and Pharmacy of Rouen, UNIROUEN, EA4108 LITIS Lab, QuantiF Team and INSERM CIC-CRB 1404, F-76183 Rouen, France; 8Department of Biostatistics, CHU Rouen, F-76000 Rouen, France; andre.gillibert@chu-rouen.fr

**Keywords:** non-small cell lung cancer, long-term survival, disease-free survival, minimally invasive surgery, VATS, RATS

## Abstract

**Simple Summary:**

Video-assisted thoracoscopic surgery (VATS) and robotic-assisted thoracoscopic surgery (RATS) are known to be safe and efficient surgical procedures to treat lung cancer. Both VATS and RATS allow anatomical resection associated with radical lymph node dissection. However, RATS, unlike VATS, allows the thoracic surgeon to mimic an open approach and to perform lung resection. We hypothesized that the technical advantages of RATS, compared with VATS, would allow more precise resection, with “better lymph node dissection” which could increase survival compared to VATS. Nevertheless, VATS, and RATS nodal up-staging are still debated, with conflicting results and in our study, as well as in the medical literature, RATS failed to show its superiority over VATS in resectable non-small cell lung cancer.

**Abstract:**

Background: Nowadays, video-assisted thoracoscopic surgery (VATS) and robotic-assisted thoracoscopic surgery (RATS) are known to be safe and efficient surgical procedures to treat early-stage non-small cell lung cancer (NSCLC). We assessed whether RATS increased disease-free survival (DFS) compared with VATS for lobectomy and segmentectomy. Methods: This retrospective cohort study included patients treated for resectable NSCLC performed by RATS or VATS, in our tertiary care center from 2012 to 2019. Patients’ data were prospectively recorded and reviewed in the French EPITHOR database. Primary outcomes were 5-year DFS for lobectomy and 3-year DFS for segmentectomy, compared by propensity-score adjusted difference of Kaplan–Meier estimates. Results: Among 844 lung resections, 436 VATS and 234 RATS lobectomies and 46 VATS and 128 RATS segmentectomies were performed. For lobectomy, the adjusted 5-year DFS was 60.9% (95% confidence interval (CI) 52.9–68.8%) for VATS and 52.7% (95%CI 41.7–63.7%) for RATS, with a difference estimated at −8.3% (−22.2–+4.9%, *p* = 0.24). For segmentectomy, the adjusted 3-year DFS was 84.6% (95%CI 69.8–99.0%) for VATS and 72.9% (95%CI 50.6–92.4%) for RATS, with a difference estimated at −11.7% (−38.7–+7.8%, *p* = 0.21). Conclusions: RATS failed to show its superiority over VATS for resectable NSCLC.

## 1. Introduction

Anatomical resection associated with lymph node dissection is the cornerstone of resectable non-small cell lung cancer (NSCLC) treatment [1]. 

Today, video-assisted thoracoscopic surgery (VATS) and robotic-assisted thoracoscopic surgery (RATS) are indicated (level of recommendation II, grade A) for the resection of early-stage NSCLC, clinical stage I [2,3], because their efficacy and safety have been proven. Compared to thoracotomy, VATS lung resection [4,5,6,7,8,9,10,11] in two randomized controlled trials [7,12,13] or RATS lung resection [11,14,15,16,17,18,19] led to better short-term outcomes, fewer adverse events, shorter hospital stays, and lower morbidity and mortality rates. Regarding short-term outcomes, the superiority of VATS or RATS is still debated in one randomized controlled trial [20] and in systematic reviews and meta-analysis and propensity-matched cohorts [11,14,15,18,21,22,23,24]. 

For long-term outcomes, overall survival (OS) and disease-free survival (DFS) are the main criteria of oncological quality to evaluate the resection performed for all cancers. No difference was reported when a minimally invasive approach, such as VATS or RATS, was compared with open surgery [11,14,15,18,21,22,23,24]. More than enhanced recovery, VATS and RATS also preserve long-term survival. 

Today, few reports [11,14,21,22,23,25,26,27] have compared the long-term survival of RATS, a recent surgical approach, with VATS, a mature, controlled, and well-known approach, for resectable lung cancers. Moreover, no large-scale randomized controlled trial has been done to evaluate the equivalence or superiority of RATS over VATS.

Our objective was to assess whether RATS led to increased 5-year DFS compared with VATS for segmentectomy and lobectomy.

## 2. Materials and Methods

### 2.1. Study Type

We conducted an observational, retrospective, and comparative cohort study, in the department of general and thoracic surgery of Rouen University Hospital, France.

### 2.2. Inclusion/Exclusion Criteria

We included all patients aged ≥18 years old, who had undergone a VATS or RATS lobectomy or segmentectomy with curative intent for a pathological NSCLC of any clinical stage, between 1 January 2012 and 31 December 2019 in our center. Bilobectomy, pneumonectomy, and histologically invalidated NSCLC were excluded. 

All cases were discussed in multidisciplinary meeting, in accordance with guidelines, during which the cancer treatment (radiotherapy, chemotherapy, surgery) was chosen. The choice between VATS and RATS was at the discretion of the surgeon and depended on the surgeon’s preference and availability of the robot: 6 days a month on the study period.

We did not exclude patients operated for a second NSCLC (multiple inclusions allowed), patients previously treated for another cancer, or patients with a clinical stage IV NSCLC with a resectable lung lesion associated with curable metastasis.

### 2.3. Surgical Procedures

VATS was performed by the modified anterior fissureless approach as described by Hansen et al. [28], using the da Vinci Si platform [29] (Intuitive Surgical, Sunnyvale, CA, USA) from 2012 to July 2018, and then the da Vinci X platform from 2018 to 2019. We first used the modified 3-arm technique with three robotic ports and an assistant port, then the modified 4-arm technique on the da Vinci Si platform without the robotic stapler, and then we switched to the da Vinci X platform with the robotic stapler.

From 2015 onwards, we used a multimodality and multidisciplinary approach [30,31,32,33] (Figure 1) combining 3D lung reconstruction (Visible Patient, Strasbourg, France) and lung tumor dye marking [32]. For RATS segmentectomy, we used near-infrared fluorescence with indocyanine green (ICG) to detect the tumor [30,34] and the intersegmental plane [35]; for VATS, we did not have a laser and did not use fluorescence.

Per- and post-operative management were guided by the principles of the enhanced recovery after surgery (ERAS) [36] program. Thoracic drainage was ensured by one chest tube that was removed as soon as possible when there were no air leaks and the amount of drained pleural effusion was less than 300 mL per day.

### 2.4. Data Collection

Patients were informed of the registration of their data in the French EPITHOR database and gave oral consent to participate in observational research projects. All data were prospectively entered in this database with a high completeness for peri-operative data in the thoracic surgery department. This database was completed and regularly checked by a dedicated data manager. The project protocol was designed in 2020, after most of the data was recorded (2012–2020) and consolidated (2019–2020). 

### 2.5. Surveillance and Follow-Up

Follow-up of thoracic surgeons was recorded in EPITHOR, but patients moving to other centers or followed by pneumologists were quickly lost to follow-up (less than three months of follow-up). Long-term follow-up data was retrospectively completed from 2019 to 2020 via our center’s medical records, letters, and telephone calls to physicians or patients if needed. All-cause mortality was also assessed by examining the French national comprehensive public register of deaths. Only medically confirmed relapses were taken into account.

Patients were considered lost to follow-up, after consolidation of data, if they had a ratio of more than 0.5 between the time without news and the time with news, they had given no news for at least 6 months, were not dead, and had no relapse at last news.

### 2.6. Outcomes

The primary outcome was 5-year DFS, expressed as a percentage. We hypothesized the superiority of RATS over VATS in both segmentectomy and lobectomy. Secondary outcomes were OS and time to relapse (TTR). Due to an unexpectedly low number of patients followed for 5 years, the 5-year DFS could not be reliably estimated in the segmentectomy subgroup; therefore, the primary analysis of segmentectomy was restricted to 3 years. The DFS was defined as the time to death from any cause or recurrence of the same lung cancer, whichever came first, with censorship at last follow-up (administrative censoring and censoring of losses to follow-up). OS was defined as the time to death from any cause with censorship at last follow-up. TTR was defined as the time to recurrence of the same lung cancer or death caused by the lung cancer, whichever came first, censored at the time of death for patients who died of another cause and at last follow-up for alive patients with no recurrence of the lung cancer. The 5-year DFS, 5-year OS and 5-year TTR were defined as proportions of patients whose, respectively, DFS, OR, and TTR were longer than 5 years.

### 2.7. TNM Staging

For consistency of TNM stage between patients, we used the 7th Ed. of lung cancer TNM [37]. We converted the 8th Ed. of lung cancer TNM to the 7th Ed. if needed, using the summary of histological reports.

### 2.8. Statistical Analysis

The consolidated data were exported from EPITHOR and analyzed in R (version 4.0) statistical software. The primary analysis was the comparison of 5-year DFS with propensity-score adjustment performed in lobectomy (first test) and in segmentectomy (second test), each at the 5% significance level without multiple testing procedure. The difference of 5-year DFS was calculated from 5-year Kaplan–Meier estimates with propensity-score weighting, using overlap weights [38], combined with multiple imputation by fully conditional specification and non-parametric percentile bootstrap in a Boot-MI sequence [39]. The multiple imputation was based on predictive mean matching (PMM) for diffusing capacity of the lung carbon monoxide (DLCO) (n = 361/844, 42.8% of missing data) and forced expiratory volume in one second (FEV1) (n = 81/844, 9.6% of missing data); no other variable had missing data. The propensity score was based on a logistic regression explaining the probability of having RATS by age (<65 years, 65–74, 75–84, ≥ 85 years), smoking status (never, former, current), Eastern Cooperative Oncology Group (ECOG) performance status (linear effect), FEV1 (linear effect), DLCO (linear effect), the preoperative clinical tumor nodes metastasis (cTNM) classification (as categorical variable), histological type, and surgeon. These covariables were chosen a priori as possible confounders by indication. The same procedure was used to compare OS and TTR. The unadjusted 30- and 90-day mortality rates were estimated by the beta product confidence procedure (BPCP) [40] and compared by melded confidence intervals with the ‘bpcp’ R package. Comparisons of baseline characteristics between groups were performed by Fisher’s exact tests for qualitative variables, Student’s *t*-tests for means, and Mann–Whitney’s test for medians.

## 3. Results

### 3.1. Description and Comparison of Baseline Characteristics

From 1 January 2012 to 31 December 2019, we performed 1159 major lung resections by minimally invasive or open approach intending to treat a confirmed or a suspected resectable NSCLC. We included 815 patients for whom 844 minimally invasive lung resections were performed, including 670 lobectomies with 234 (34.9%) RATS and 436 (65.1%) VATS and 174 segmentectomies with 128 (73.5%) RATS and 46 (26.5%) VATS for a histologically confirmed NSCLC (Figure 2).

These 844 procedures were performed by seven surgeons. Two surgeons (#1: *n* = 290; #2: *n* = 64) performed 354 (97.8%) RATS procedures. Four surgeons (#1: *n* = 144; #2: *n* = 156, #3: *n* = 110; #4: *n* = 53) performed 463 (96.1%) VATS procedures. Baseline characteristics of patients with lobectomy and segmentectomy are shown in Table 1.

### 3.2. Quality of Follow-Up

A total of 48 (5.8%) patients were lost to follow-up. In patients who had lobectomy by VATS or RATS, respectively, 28 (6.4%) and 15 (6.4%) were lost to follow-up, and in the segmentectomy group, respectively, 3 (6.5%) and 2 (1.6%) were lost to follow-up. After exclusion of patients who died and patients with recurrence of lung cancer, the overall median duration of oncological follow-up, capped to 60 months, in VATS and RATS patients was, respectively, 23.9 months (interquartile range (IQR) 7.0–53.8) and 23.4 months (IQR 11.0–42.5) without significant difference (*p* = 0.96). 

### 3.3. Primary Analysis

After propensity-score weighting, the 5-year DFS of patients who had lobectomy by VATS and RATS, was estimated at 60.9% (95% CI 52.9–68.8%) and 52.7% (95% CI 41.7–63.7%), respectively, with a difference of −8.3% (−22.2 to +4.9%, *p* = 0.24, first primary analysis). After propensity-score weighting, the 5-year DFS (planned primary analysis) of patients who had segmentectomy by VATS and RATS could not be estimated, but the 3-year DFS was estimated at 84.6% (95% CI 69.8–99.0%) and 72.9% (95% CI 50.6–92.4%), respectively, with a difference of −11.7% (95% CI −38.7 to +7.8%, *p* = 0.21) (Table 2 and Figure 3).

### 3.4. Per-Operative and Short-Term Post-Operative Outcomes

The unadjusted frequency of nodal up-staging, conversion to thoracotomy, 30 and 90-day mortality rates, mean hospital length of stay, and complication rates and stages were not significantly different between VATS and RATS groups (Table 3) except for conversions to thoracotomy during a segmentectomy (*p* = 0.04). 

The operative time was significantly shorter for RATS segmentectomy than for VATS segmentectomy. After a post hoc adjustment for the surgeon in a general linear model, the difference of mean duration for RATS segmentectomy was estimated at +5.8 min for RATS (95% CI −9.3 to +21.0, *p* = 0.45) compared with VATS. Indeed, the most experienced surgeon performed 78.9% (n = 101/128) of all RATS segmentectomies but only 13.0% (n = 6/46) of all VATS segmentectomies.

### 3.5. Characteristics of Recurrences

Long-term tumor-related outcomes and recurrences are described in Table 4. Recurrences and treatments were not significantly different between VATS and RATS groups, and 79.1% of patients were free of recurrence in the lobectomy group and 85.0% in the segmentectomy group. Among recurrences occurring within 5 years of the VATS or RATS lobectomy or segmentectomy, 63 (39.9%) were local, 59 (37.3%) were metastatic, and 36 (22.8%) were both local and metastatic recurrences. Between 71.3 and 86.7% of deaths occurring beyond 90 days, were attributed to the operated lung cancer among RATS-VATS and lobectomy-segmentectomy subgroups. A secondary planned subgroup analysis (Table 5) of stage IA tumors found non-significant DFS, OS, and TTR differences, between RATS and VATS.

## 4. Discussion

### 4.1. Summary of Main Results

In our cohort, we failed to show the long-term oncological superiority of RATS over VATS for resectable lung cancer treated by lobectomy or segmentectomy and lymph node dissection, with a 5-year adjusted estimated difference of DFS of −8.3% (−22.2 to +4.9%, *p* = 0.24) for lobectomy and a 3-year adjusted difference of DFS of −11.7% (95% CI −38.7 to +7.8%, *p* = 0.21) for segmentectomy.

### 4.2. Comparison with Literature

In comparison with systematic review and meta-analysis [11,14,21,22,23], and matched cohort analysis [25,26] (Table 6) that analyzed long-term survival data after comparing VATS and RATS approaches, our results confirm the absence of major superiority of the robotic approach.

Nevertheless, we can cite two articles that seem to show an advantage of the robotic approach in terms of recurrence rate and DFS. First of all, the meta-analysis of Ma et al. [21], which showed an advantage of RATS for crude recurrence rate (odds ratio (OR): 0.53; 95% CI 0.37–0.74, *p* < 0.001) for lobectomy but not for segmentectomy, *p* = 0.18. However, it did not take into account the difference in length of follow-up attributed to the fact that RATS procedures are usually more recent than VATS. Moreover, Ma et al. [21] found no advantage of RATS for 5-year DFS or OS. Next, in the meta-analysis of Wu et al. [23] an advantage was shown for the robotic approach, compared with VATS, for lobectomy, for 5-year DFS (hazard ratio (HR): 0.76; 95% CI: 0.59–0.97, *p* = 0.03, but without significant superiority of RATS for 5-year OS (HR: 0.77; 95% CI: 0.57–1.05, *p* = 0.10). Compared with larger VATS series [8,41,42,43], RATS series [15,18,19,44,45,46], systematic reviews and meta-analyses [11,14,21,22,23], and matched cohort analyses [25,26,27] (Table 6), our long-term survival rates for lobectomy are consistent, with 5-year DFS rates of 60.9% for VATS and 52.7% for RATS (propensity score adjusted), *p* = 0.24, and 5-year OS rates of 76.0% for VATS and 70.4% for RATS, *p* = 0.54 (propensity score adjusted).

Our multidisciplinary approach [30,31,32,33] (Figure 1 and Figure 3) allows a minimally invasive tailored anatomical segmentectomy with improved surgical margins and oncological effectiveness and safety with preserved long-term survival, a 3-year DFS of 82.4% for VATS, and 78.0% for RATS, *p* = 0.59 (propensity score adjusted) and a 3-year OS of 87.8% for VATS, and 90.1% for RATS, *p* = 0.81 (propensity score adjusted). Our long-term survival results are consistent with the literature [21,33,47,48].

Compared with our VATS experience with the anterior fissureless approach technique [28], RATS allows the thoracic surgeon to mimic an open approach to perform lung resection. However, regardless of the approach, the lung resection remains the same, with anatomical resection associated with radical lymph node dissection. Thus, long-term survival in our report and in the literature [11,14,21,22,23,25,26,27], does not seem to be greatly influenced by the surgical approach. We hypothesized that the technical advantages of RATS, compared with VATS, would allow more precise resection, with “better lymph node dissection” which could allow increased survival compared with VATS. Nevertheless, VATS, and RATS nodal up-staging compared with thoracotomy are still debated, with conflicting results about lower upstaging by VATS [11,27,49], higher upstaging by RATS [11,27], or a lack of difference [14,22,23,26], but without a significant impact on long-term survival. In our cohort we did not find significant differences in nodal up-staging rates in VATS and RATS groups; however, the sample size was not large enough to conclude that the staging rates were equivalent. Finally, even if lymph node dissection seemed easier to complete, it is more operator dependent than approach dependent, all three approaches (open surgery, VATS, RATS) allowing a complete quality oncological resection. This could be one of the reasons explaining the lack of significant difference of OS and DFS between RATS and VATS.

### 4.3. Strengths

Our cohort is one of the largest comparing 5-year outcomes between RATS and VATS, and although underpowered, will add high-quality data to the meta-analysis. Our database is prospectively completed and regularly controlled by our dedicated data manager, guaranteeing a good quality of data.

Although the study was not randomized, the surgical indications for VATS and RATS are mostly the same in our thoracic surgery department, except our preference for RATS for segmentectomy, in a multimodal [30,31,32] approach but lobectomy and segmentectomy were analyzed separately.

### 4.4. Limitations

We report one of the largest single center surgical cohorts of lobectomy and segmentectomy performed by VATS and RATS. However, the cohort is still too small to draw any firm conclusions regarding the long-term oncological outcomes of VATS and RATS. For 5-year OS the difference between thoracotomy and VATS or RATS is less than 5% according to the largest comparative studies and meta-analyses [11,14,21,22,23,25,26,27].

Another limitation of this study is the follow-up period which was shorter than planned due to the COVID-19 pandemic disrupting our surgical activity. We have now adapted our surgical activity to the pandemic, and a new study could be conducted with longer follow-up which would perhaps allow us to answer our initial research question.

As this study is observational, confounding by indication was possible. The indication of VATS or RATS mainly depended on the surgeon’s preferences and the type of surgery, with a preference for RATS for segmentectomy and for VATS for lobectomy. Since segmentectomy and lobectomy were analyzed separately and the surgeon was included in the propensity score, these main indication biases were canceled. However, we noticed a significant difference of ECOG performance status between RATS and VATS in the lobectomy group suggesting that other confounders were possible. There were adjustments on main prognostic factors, including the ECOG performance status, but a residual confounding bias is possible. However, tumor stages were not significantly different between groups, and propensity-score adjustment had no major effect on DFS and OS differences between VATS and RATS, suggesting that the indication bias may not have had a major impact on results.

The anterior VATS approach was introduced in our department in 2008, and RATS in 2012. Our cohort includes our RATS learning curve, but not that of VATS, which may reduce the comparability of procedures.

### 4.5. Perspectives

Following international recommendations [3,50,51,52] we used minimally invasive procedures to perform major lung resection for most resectable NSCLC, both for early stages and for advanced cases. Our results regarding short- and long-term survival are encouraging for these “extended” indications of minimally invasive lobectomy, with few conversions to thoracotomy, few postoperative complications, and preserved long-term survival in comparison with the literature [11,14,21,22,23,25,26,27]. 

For early-stage NSCLC, the time may no longer be ripe for the opposition and confrontation of VATS and RATS, because both techniques can be used by the same surgical team making it possible to optimize the oncological management of patients while also taking into account the logistical and economic constraints [29,53,54] in our hospitals.

We believe that evidence will emerge in the next few years to support robotic surgery as the optimal minimally invasive platform for complex lung resections as segmentectomy and locally advanced NSCLC, as RATS allows surgeons to mimic open surgery while maintaining the advantages of minimally invasive approaches [1].

## 5. Conclusions

We failed to show the long-term oncological superiority of RATS over VATS in a single center cohort in real-life clinical practice. However, our cohort was too small to detect moderate differences between RATS and VATS. Perhaps the main reason is that both VATS and RATS allow the surgeon to perform oncological resection with complete lymph node dissection and the main limitation is not the tool but the operator. Nevertheless, we plan to compare RATS with VATS in a future multicenter cohort of the French EPITHOR database, once the duration of follow-up is improved.

## Figures and Tables

**Figure 1 cancers-14-02611-f001:**
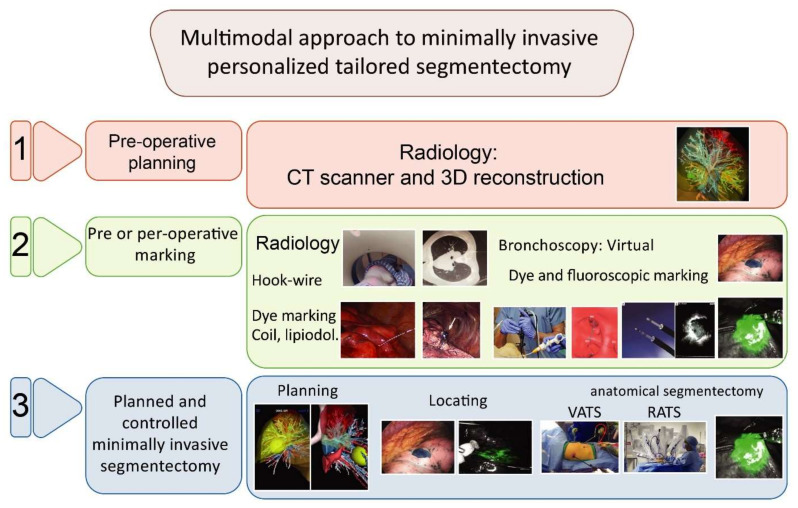
Multimodal approach to minimally invasive personalized tailored segmentectomy in 3 steps.

**Figure 2 cancers-14-02611-f002:**
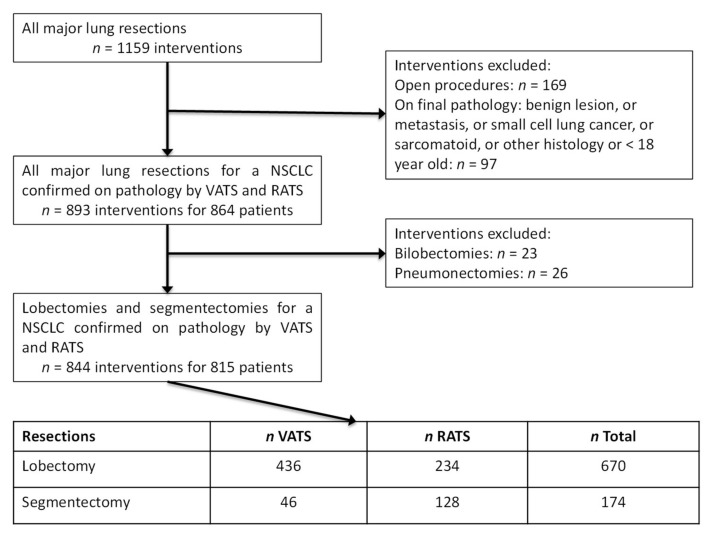
Flow-chart of our surgical series of resectable NSCLC treated by lobectomy and segmentectomy performed by VATS and RATS from the 1st January 2012 to 31st December 2019.

**Figure 3 cancers-14-02611-f003:**
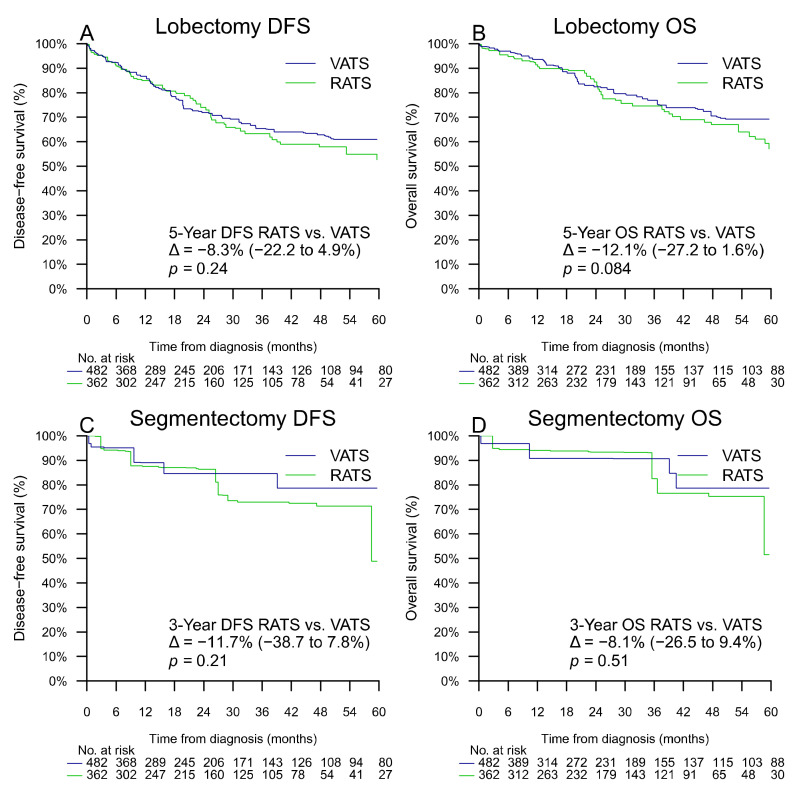
Propensity-score adjusted DFS for lobectomy (**A**), OS for lobectomy (**B**), DFS for segmentectomy (**C**), and OS for segmentectomy (**D**). Δ represents the difference of percentage of survival with its 95% confidence interval.

**Table 1 cancers-14-02611-t001:** Pre-operative characteristics of patients who had lobectomy and segmentectomy, according to VATS and RATS procedure.

	Lobectomy			Segmentectomy		
	VATS *n* = 436	RATS *n* = 234	*p* *	VATS *n* = 46	RATS *n* = 128	*p* *
**Age, year, mean ± SD**	65.24 ± 9.36	64.49 ± 10.49	0.35	63.29 ± 8.13	64.34 ± 8.24	0.46
**Gender, Female *n* (%)**	139 (31.9%)	87 (37.2%)	0.20	19 (41.3%)	53 (41.4%)	1.00
**Smoking status, *n* (%)**			0.11			0.087
**Never**	113 (25.9%)	71 (30.3%)		8 (17.4%)	34 (26.6%)	
**Former**	139 (31.9%)	57 (24.4%)		14 (30.4%)	20 (15.6%)	
**Current**	184 (42.2%)	106 (45.3%)		24 (52.2%)	74 (57.8%)	
**Pulmonary co-morbidities, *n* (%)**						
**COPD**	99 (22.7%)	48 (20.5%)	0.58	11 (23.9%)	21 (16.4%)	0.36
**Emphysema**	6 (1.4%)	2 (0.9%)	0.86	2 (4.3%)	3 (2.3%)	0.80
**Sleep apnea**	27 (6.2%)	9 (3.8%)	0.27	1 (2.2%)	7 (5.5%)	0.65
**Prior thoracic surgery**	44 (10.1%)	13 (5.6%)	0.057	6 (13%)	27 (21.1%)	0.33
**History of treated cancer, *n* (%)**	130 (29.8%)	63 (26.9%)	0.49	21 (45.7%)	54 (42.2%)	0.81
**Lung cancer**	18 (4.1%)	5 (2.1%)	0.26	7 (15.2%)	18 (14.1%)	1.00
**Cardiovascular co-morbidities, *n* (%)**						
**High blood pressure**	117 (26.8%)	70 (29.9%)	0.45	15 (32.6%)	30 (23.4%)	0.31
**Coronary artery disease**	42 (9.6%)	14 (6%)	0.13	3 (6.5%)	3 (2.3%)	0.38
**Cardiac arrythmia**	26 (6%)	9 (3.8%)	0.32	0 (0%)	7 (5.5%)	0.22
**Stroke**	15 (3.4%)	7 (3%)	0.95	2 (4.3%)	5 (3.9%)	1.00
**Pre-operative treatment, *n* (%)**						
**Immunotherapy**	4 (0.9%)	2 (0.9%)	1.00	0 (0%)	2 (1.6%)	1.00
**Corticosteroid therapy**	5 (1.1%)	1 (0.4%)	0.64	0 (0%)	0 (0%)	1.00
**Immunosuppressive therapy**	13 (3%)	2 (0.9%)	0.12	0 (0%)	2 (1.6%)	1.00
**Functional PFT**						
**FEV1, %, mean ± SD**	85.42 ± 18.39	85.3 ± 19.85	0.94	87.17 ± 21.16	88.55 ± 20.19	0.71
**FEV1 Missing data**	33 (7.6%)	32 (13.7%)	0.018	4 (8.7%)	12 (9.4%)	1.00
**DLCO, %, mean ± SD**	72.99 ± 18.46	74.29 ± 19.1	0.54	75.23 ± 18.75	70.75 ± 15.94	0.24
**DLCO Missing data**	139 (31.9%)	131 (56%)	<0.0001	11 (23.9%)	80 (62.5%)	<0.0001
**ASA score**			0.45			0.22
**1**	106 (24.3%)	56 (23.9%)		16 (34.8%)	27 (21.1%)	
**2**	210 (48.2%)	119 (50.9%)		21 (45.7%)	76 (59.4%)	
**3**	115 (26.4%)	59 (25.2%)		9 (19.6%)	24 (18.8%)	
**4**	5 (1.1%)	0 (0%)		0 (0%)	1 (0.8%)	
**ECOG Performance status**			<0.0001			0.77
0	231 (53%)	180 (76.9%)		32 (69.6%)	93 (72.7%)	
1	186 (42.7%)	49 (20.9%)		14 (30.4%)	33 (25.8%)	
≥2	19 (4.4%)	5 (2.1%)		0 (0%)	2 (1.6%)	
**Charlson Index, mean ± SD**	3.59 ± 2.05	3.33 ± 1.86	0.11	3.70 ± 2.14	3.48 ± 1.93	0.52
**Primary tumor location, *n* (%)**			0.29			0.001
**RUL**	61 (14.3%)	23 (10.6%)		8 (19.5%)	15 (14.9%)	
**RML**	115 (26.9%)	50 (23%)		15 (36.6%)	44 (43.6%)	
**RIL**	64 (15%)	34 (15.7%)		16 (39%)	15 (14.9%)	
**LUL**	21 (4.9%)	17 (7.8%)		0 (0%)	0 (0%)	
**LIL**	167 (39%)	93 (42.9%)		2 (4.9%)	27 (26.7%)	
**Missing data**	8/436 (1.8%)	17/234 (7.3%)		5/46 (10.9%)	27/128 (21.1%)	
**Pre-operative stage #, *n* (%)**			0.65			0.43
**0**	0 (0%)	1 (0.4%)		0 (0%)	0 (0%)	
**IA**	213 (48.9%)	112 (47.9%)		38 (82.6%)	111 (86.7%)	
**IB**	103 (23.6%)	50 (21.4%)		2 (4.3%)	8 (6.2%)	
**IIA**	35 (8%)	27 (11.5%)		0 (0%)	0 (0%)	
**IIB**	32 (7.3%)	14 (6%)		1 (2.2%)	4 (3.1%)	
**IIIA**	36 (8.3%)	20 (8.5%)		3 (6.5%)	2 (1.6%)	
**IIIB**	1 (0.2%)	0 (0%)		0 (0%)	0 (0%)	
**IV**	16 (3.7%)	10 (4.3%)		2 (4.3%)	3 (2.3%)	
**Histology, *n* (%)**			0.005			0.49
**Adenocarcinoma**	296 (67.9%)	163 (69.7%)		37 (80.4%)	106 (82.8%)	
**Squamous cell carcinoma**	97 (22.2%)	44 (18.8%)		3 (6.5%)	10 (7.8%)	
**Typical and atypical carcinoid tumor**	10 (2.3%)	18 (7.7%)		2 (4.3%)	8 (6.2%)	
**Large cell carcinoma**	10 (2.3%)	4 (1.7%)		3 (6.5%)	2 (1.6%)	
**Others**	23 (5.3%)	5 (2.1%)		1 (2.2%)	2 (1.6%)	

* *p*-values comparing VATS to RATS without adjustment; ASA: American Society of Anesthesiologists; COPD: chronic obstructive pulmonary disease; DLCO: diffusing capacity of the lung carbon monoxide; ECOG performance status: Eastern Cooperative Oncology Group performance status: FEV1: forced expiratory volume in one second; LIL: left lower lobe; LUL: left upper lobe; PFT: pulmonary functionary test; RATS: robotic-assisted thoracoscopic surgery; VATS: video-assisted thoracoscopic surgery; RIL: right inferior lobe; ML: middle lobe; RUL: right upper lobe; SD: standard deviation; **#**, According 7th Ed. of lung cancer TNM.

**Table 2 cancers-14-02611-t002:** Long-term survival results of patients who had lobectomy and segmentectomy, according to VATS and RATS procedure.

	VATS	RATS		
Lobectomy				
Sample size	***n* events/N patients**	***n* events/N patients**		
5-Y DFS	137/436	62/234		
5-Y OS	103/436	46/234		
5-Y TTR	111/436	53/234		
Unadjusted	**Surv (95% CI)**	**Surv (95% CI)**	**Surv difference (95% CI)**	** *p* **
5-Y DFS	53.9% (47.7–60.2%)	57.4% (47.2–67.4%)	3.6% (−8.9 to 15.4%)	0.56
5-Y OS	61.2% (54.9–67.6%)	60.6% (48.8–71.6%)	−0.7% (−13.9 to 12.1%)	0.92
5-Y TTR	61.8% (55.5–68.0%)	65.4% (56.2–73.8%)	3.6% (−7.5 to 14.2%)	0.51
Propensity score adjusted	**Surv (95% CI)**	**Surv (95% CI)**	**Surv difference (95% CI)**	** *p* **
5-Y DFS	60.9% (52.9–68.8%)	52.7% (41.7–63.7%)	−8.3% (−22.2 to 4.9%)	0.24
5-Y OS	69.3% (61.5–77.4%)	57.2% (45.2–68.9%)	−12.1% (−27.2 to 1.6%)	0.084
5-Y TTR	66.2% (58.5–73.9%)	60.2% (49.6–70.4%)	−6.0% (−19.3 to 6.5%)	0.37
Segmentectomy				
Sample size	***n* events/N patients**	***n* events/N patients**		
3-Y DFS	6/46	18/128		
3-Y OS	3/46	9/128		
3-Y TTR	5/46	18/128		
Unadjusted	**Surv (95% CI)**	**Surv (95% CI)**	**Surv difference (95% CI)**	** *p* **
3-Y DFS	82.8% (68.9–94.7%)	77.4% (67.2–86.9%)	−5.4% (−21.3 to 11.8%)	0.50
3-Y OS	89.3% (75.0–100%)	87.3% (77.9–94.9%)	−2.0% (−16.6 to 14.4%)	0.76
3-Y TTR	84.7% (70.6–96.6%)	77.4% (67.2–86.9%)	−7.3% (−23.2 to 9.8%)	0.37
Propensity score adjusted	**Surv (95% CI)**	**Surv (95% CI)**	**Surv difference (95% CI)**	** *p* **
3-Y DFS	84.6% (69.8–99.0%)	72.9% (50.6–92.4%)	−11.7% (−38.7 to 7.8%)	0.21
3-Y OS	90.7% (79.1–100%)	82.6% (65.1–99.9%)	−8.1% (−26.5 to 9.4%)	0.51
3-Y TTR	87.4% (73.8–100%)	72.9% (50.6–92.4%)	−14.4% (−41.5 to 4.0%)	0.12

3-Y: 3 year; 5-Y: 5 year; DFS: disease free survival; OS: overall survival; TTR: time to relapse (censorship on cancer-unrelated death); Surv: any type of survival; RATS: robotic-assisted thoracoscopic surgery; VATS: video-assisted thoracoscopic surgery. CI: confidence intervals.

**Table 3 cancers-14-02611-t003:** Per-operative and post-operative characteristics of patients and tumors.

	Lobectomy			Segmentectomy		
	VATS *n* = 436	RATS *n* = 234	*p*	VATS *n* = 46	RATS *n* = 128	*p*
Conversion to thoracotomy, *n* (%)						
Total	48 (11%)	16 (6.8%)	0.10	5 (10.9%)	3 (2.3%)	0.062
For operative complications	18 (4.1%)	4 (1.7%)	0.056	1 (2.2%)	1 (0.8%)	0.04
For disease reasons	11 (2.5%)	3 (1.3%)		2 (4.3%)	0 (0%)	
Due to symphysis and fissure	13 (3%)	2 (0.9%)		1 (2.2%)	1 (0.8%)	
For other reasons	6 (1.4%)	7 (3%)		1 (2.2%)	1 (0.8%)	
Operative time (min), med (Q1; Q3)	150 (120; 180)	150 (110; 180)	0.09 *	150 (120; 180)	100 (84; 131)	<0.0001 *
Clavien–Dindo complications			0.26			0.079
None	232 (53.2%)	141 (60.3%)		31 (67.4%)	101 (78.9%)	
I	58 (13.3%)	21 (9%)		6 (13%)	5 (3.9%)	
II	102 (23.4%)	55 (23.5%)		7 (15.2%)	20 (15.6%)	
IIIa	13 (3%)	2 (0.9%)		0 (0%)	1 (0.8%)	
IIIb	23 (5.3%)	11 (4.7%)		1 (2.2%)	1 (0.8%)	
IVa	1 (0.2%)	0 (0%)		0 (0%)	0 (0%)	
IVb	0 (0%)	0 (0%)		0 (0%)	0 (0%)	
V	7 (1.6%)	4 (1.7%)		1 (2.2%)	0 (0%)	
Mean ± SD (from 0 to 7)	1.03 ± 1.41	0.89 ± 1.38	0.24	0.67 ± 1.32	0.41 ± 0.84	0.12
Length of stay, day, median (Q1; Q3)	5 (4; 8)	5 (4; 7)	0.09 *	4 (3; 5.8)	4 (3; 5)	0.84 *
Re-admission, *n*,%	16 (3.7%)	10 (4.3%)	0.85	2 (4.3%)	0 (0%)	0.14
Infection	6 (31.6%)	4 (30.8%)		0	0	
Pleural effusion	2 (10.5%)	6 (46.2%)		0	0	
Hemorrhage	2 (10.5%)	1 (7.7%)		0	0	
Pulmonary failure	2 (10.5%)	0 (0%)		0	0	
Thromboembolic complication	2 (10.5%)	0 (0%)		0	0	
Other	5 (26.3%)	2 (15.4%)		2 (100%)	0	
Mortality						
At day 30	7 (1.61%)	4 (1.73%)	1.00 †	1 (2.17%)	0 (0%)	0.53 †
At day 90	11 (2.58%)	6 (2.63%)	1.00 †	1 (2.17%)	1 (0.81%)	0.95 †
Pathologic stage #, *n* (%)			0.29			0.37
0	8 (1.8%)	4 (1.7%)		1 (2.2%)	5 (3.9%)	
IA	134 (30.7%)	78 (33.3%)		30 (65.2%)	86 (67.2%)	
IB	145 (33.3%)	61 (26.1%)		6 (13%)	18 (14.1%)	
IIA	49 (11.2%)	32 (13.7%)		3 (6.5%)	5 (3.9%)	
IIB	41 (9.4%)	19 (8.1%)		2 (4.3%)	9 (7%)	
IIIA	42 (9.6%)	34 (14.5%)		4 (8.7%)	2 (1.6%)	
IIIB	3 (0.7%)	2 (0.9%)		0 (0%)	0 (0%)	
IV	14 (3.2%)	4 (1.7%)		0 (0%)	3 (2.3%)	
Nodal Up-staging, *n* (%)						
cN0 → pN+ (N1 and/or N2)	53 (12.2%)	29 (12.4%)	1.00	3 (6.5%)	7 (5.5%)	1.00
cN0 → pN1	37 (8.5%)	18 (7.7%)	0.84	2 (4.3%)	6 (4.7%)	1.00
cN0 → pN2	16 (3.7%)	11 (4.7%)	0.65	1 (2.2%)	1 (0.8%)	0.92
Adjuvant therapy, *n*, %			0.17			0.18
Chemotherapy	89 (20.4%)	41 (17.5%)		9 (19.6%)	13 (10.2%)	
Radiotherapy	5 (1.1%)	6 (2.6%)		0 (0%)	0 (0%)	
Chemotherapy and Radiotherapy	8 (1.8%)	9 (3.8%)		0 (0%)	2 (1.6%)	
Refused by the patient	17 (3.9%)	7 (3%)	0.71	0 (0%)	1 (0.8%)	1.00
Adjuvant therapy by node status, *n*, %						
pN+	51 (54.8%)	34 (56.7%)	0.96	3 (75%)	6 (85.7%)	1.00
pN1	32 (52.5%)	13 (43.3%)	0.55	2 (66.7%)	5 (83.3%)	1.00
pN2	19 (59.4%)	21 (70%)	0.54	1 (100%)	1 (100%)	1.00

† Percentage estimates by Kaplan–Meier and comparison by beta product confidence procedure; **#**: According AJJC 7th ed.; * Mann–Whitney test; min: minute; Q: quartile; RATS: robotic-assisted thoracoscopic surgery; SD: standard deviation; VATS: video-assisted thoracoscopic surgery.

**Table 4 cancers-14-02611-t004:** Long-term tumor-related outcomes of patients with RATS or VATS lobectomy or segmentectomy.

	Lobectomy			Segmentectomy		
	VATS *n* = 436	RATS *n* = 234	*p*	VATS *n* = 46	RATS *n* = 128	*p*
Follow-up of disease-free survivors, months, median (Q1; Q3)	25.3 (6.9; 56.6)	24 (10.1; 43.2)	0.90 *	16.0 (7.2; 48.0)	22.7 (11.8; 40.9)	0.42 *
Lost to follow-up, *n*, %	28 (6.4%)	15 (6.4%)	1.00	3 (6.5%)	2 (1.6%)	0.23
First recurrence within 5-Y, *n*,%			0.52			0.75
None	349 (80.4%)	186 (79.8%)		40 (87%)	108 (84.4%)	
Local	32 (7.4%)	23 (9.9%)		2 (4.3%)	6 (4.7%)	
Metastatic	36 (8.3%)	14 (6%)		3 (6.5%)	6 (4.7%)	
Local and metastatic	17 (3.9%)	10 (4.3%)		1 (2.2%)	8 (6.2%)	
Treatment of first recurrence within 5-Y, *n*,%			0.73			0.10
Chemotherapy	26 (30.6%)	15 (31.9%)		1 (16.7%)	9 (45%)	
Radiotherapy	11 (12.9%)	9 (19.1%)		2 (33.3%)	0 (0%)	
Chemotherapy and Radiotherapy	8 (9.4%)	3 (6.4%)		1 (16.7%)	4 (20%)	
Palliative care only	40 (47.1%)	20 (42.6%)		2 (33.3%)	7 (35%)	
pTNM stage #	**5-Y Disease Free Survival, % (95% CI)**			**3-Y Disease Free Survival, % (95% CI)**		
IA	68.0% (57.8–76.3%)	69.9% (49.2–83.1%)		82.4% (57.5–94.3%)	78.0% (64.3–87.5%)	
IB	42.5% (27.9–56.6%)	63.7% (27.2–80.6%)		UTC	UTC	
IIA	65.4% (39.3–83.3%)	38.2% (2.1–64.4%)		UTC	UTC	
IIB	40.7% (7.6–69.0%)	18.1% (0.6–59.4%)		UTC	UTC	
IIIA	26.7% (4.7–53.2%)	UTC		UTC	UTC	
IIIB	UTC	UTC		UTC	UTC	
IV	7.3% (0.2–33.8%)	30.0% (1.1–70.1%)		UTC	UTC	
pTNM stage #	**5-Y Overall Survival, % (95% CI)**			**3-Y Overall Survival, % (95% CI)**		
IA	76.0% (65.8–83.7%)	70.4% (49.6–84.3%)		87.8% (62.7–97.7%)	90.1% (77.9–96.3%)	
IB	47.9% (32.2–62.4%)	69.5% (33.9–86.1%)		UTC	UTC	
IIA	63.3% (36.5–83.5%)	28.5% (1.2–64.8%)		UTC	UTC	
IIB	44.3% (12.6–72.5%)	UTC		UTC	UTC	
IIIA	36.5% (12.9–61.0%)	UTC		UTC	UTC	
IIIB	UTC	UTC		UTC	UTC	
IV	40.5% (6.6–76.0%)	48.2% (7.7–84.5%)		UTC	UTC	
Death cause, beyond day 90, *n*,%			0.82			0.60
Related to the lung cancer	77 (71.3%)	38 (77.6%)		4 (80%)	13 (86.7%)	
Related to another cancer	13 (12%)	5 (10.2%)		0 (0%)	1 (6.7%)	
Non-cancer disease	18 (16.7%)	6 (12.2%)		1 (20%)	1 (6.7%)	

* Mann–Whitney test; 3-Y: 3-year; 5-Y: 5-year; RATS: robotic-assisted thoracoscopic surgery; UTC: unable to be calculate due to the small sample size; VATS: video-assisted thoracoscopic surgery; **#** According 7th Ed. of lung cancer TNM.

**Table 5 cancers-14-02611-t005:** Unadjusted and adjusted comparison of long-term survival in the subgroup of patients with cTNM stage I tumors.

	VATS	RATS		
Lobectomy for cTNM IA#				
Sample size	***n* events/N patients**	***n* events/N patients**		
5-Y DFS	44/213	17/112		
5-Y OS	30/213	15/112		
5-Y TTR	34/213	13/112		
Unadjusted surv	**Surv (95% CI)**	**Surv (95% CI)**	**Surv difference (95% CI)**	** *p* **
5-Y DFS	68.0% (59.2–76.2%)	69.9% (54.7–83.1%)	1.9% (−16.1 to 18.0%)	0.81
5-Y OS	76.0% (67.6–84.1%)	70.4% (54.0–84.7%)	−5.7% (−24.2 to 11.1%)	0.54
5-Y TTR	74.6% (66.4–82.3%)	75.3% (60.0–88.4%)	0.6% (−16.6 to 16.3%)	0.92
Propensity score adjusted surv	**Surv (95% CI)**	**Surv (95% CI)**	**Surv difference (95% CI)**	** *p* **
5-Y DFS	71.9% (59.4–83.0%)	67.7% (50.5–84.4%)	−4.2% (−24.3 to 17.1%)	0.80
5-Y OS	80.9% (69.5–90.5%)	66.2% (47.6–84.9%)	−14.7% (−34.8 to 7.3%)	0.20
5-Y TTR	75.5% (63.3–86.1%)	74.0% (57.0–89.6%)	−1.5% (−21.5 to 18.4%)	0.99
Segmentectomy for cTNM IA #				
Sample size	**n events/N patients**	**n events/N patients**		
3-Y DFS	5/38	16/111		
3-Y OS	3/38	7/111		
3-Y TTR	4/38	16/111		
Unadjusted	**Surv (95% CI)**	**Surv (95% CI)**	**Surv difference (95% CI)**	** *p* **
3-Y DFS	82.4% (66.4–96.0%)	78.0% (66.8–87.7%)	−4.5% (−21.8 to 14.2%)	0.59
3-Y OS	87.8% (71.5–100.0%)	90.1% (81.6–96.8%)	2.3% (−12.3 to 20.2%)	0.81
3-Y TTR	84.7% (68.8–96.9%)	78.0% (66.8–87.7%)	−6.7% (−23.6 to 12.0%)	0.43
Propensity score adjusted	**Surv (95% CI)**	**Surv (95% CI)**	**Surv difference (95% CI)**	** *p* **
3-Y DFS	83.4% (67.9–100.0%)	68.3% (44.3–91.4%)	−15.1% (−44.4 to 7.5%)	0.17
3-Y OS	89.5% (77.4–100.0%)	80.8% (62.6–99.9%)	−8.7% (−28.2 to 9.9%)	0.51
3-Y TTR	86.3% (71.4–100.0%)	68.3% (44.3–91.4%)	−18.0% (−47.7 to 3.9%)	0.096

3-Y: 3-year; 5-Y: 5-year; DFS: disease free survival; OS: overall survival; Surv: any type of survival; TTR: time to relapse (censorship on cancer-unrelated death); RATS: robot-assisted thoracoscopic surgery; VATS: video-assisted thoracoscopic surgery.

**Table 6 cancers-14-02611-t006:** Main results of systematic review and meta-analysis, and matched cohort analysis included in our analysis regarding long-term survival following RATS, VATS lobectomy and segmentectomy for NSCLC.

Reference	Study Setting	OS RATS vs. VATS Adjusted HR (95% CI)	DFS RATS vs. VATS Adjusted HR (95% CI)
Ma et al. 2021; BMC Cancer; Systematic review and meta-analysis [21]	Systematic review and meta-analysis 18 studies included Lobectomy + segmentectomy 5114 RATS 6133 VATS 2008 to 2019	1.02 (0.82–1.26)	1.03 (0.66–1.61)
Aiolfi 2021 [11]	Systematic review and meta-analysis Lobectomy 34 studies included 79,171 VATS 15,390 RATS 1990 to 2018	1.53 (0.87–2.88)	
Wu 2020 [23]	Systematic review and meta-analysis Lobectomy 25 studies included 7135 RATS 43,269 VATS 2011 to 2020	0.77 (0.57–1.05)	0.76 (0.59–0.97)
Kneuertz 2020 [22]	Society of Thoracic Surgery General Thoracic Surgery Database Lobectomy stage I to III 245 RATS 118 VATS 2012 to 2017	0.72 (0.42–1.22)	0.67 (0.43–1.04)
Veluswamy 2019 [26]	SEER–Medicare database. Lobectomy stage I to IIIA Age > 65 years 338 RATS 1127 VATS 2008 to 2013	0.91 (0.70–1.18)	
Yang 2017 [27]	Retrospective single-center cohort Lobectomy stage IA to IB 172 RATS (after matching) 141 VATS (after matching) 2002 to 2012	1.07 (0.62–1.83)	1.12 (0.73–1.74)

DFS: disease free survival; OS: overall survival; HR: hazard ratio; RATS: robot-assisted thoracoscopic surgery; VATS: video-assisted thoracoscopic surgery; SEER: surveillance, epidemiology and end results.

## Data Availability

The pseudonymized data presented in this study are available on request from the corresponding author. The data are not publicly available due to legal restrictions in France.

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
