# Peer review of "Long-Term Survival Following Minimally Invasive Lung Cancer Surgery: Comparing Robotic-Assisted and Video-Assisted Surgery"

_cancers, 2022, doi:10.3390/cancers14112611_

Round 1

Reviewer 1 Report

In the present paper Authors reported on a single center large series of NSCLC patients who underwent minimally invasive Segmentectomy or lobectomy. They retrospectively compared the outcomes of VATS vs RATS approaches, in terms of disease free survival, overall survival and time to recurrence. No significant difference was observed in each study endpoint.

The study population seems to be well balanced and numbers are acceptable to pursue the study aims. However, I have a few concerns about this study methodology:

1) Although the 844 reported surgical cases were performed by seven surgeons, the majority of RATS procedures (97.8%) were performed by two of them only. Similarly, four surgeons performed 96.1% of VATS. Were the first two surgeon included in the latter four? If not, the present paper appears to be a comparison between surgeon and their personal techniques rather than between two different kind of surgical approach.

2) Page 4, line 160: TNM staging were converted to the 7th edition. I can’t understand why. Please convert to the most recent one (the 8th).

3) Page 5, lines 173-183: here the covariates used for propensity-score weighting are reported. Some important variables such as smoking history and histology are lacking. Both of them are unbalanced between study groups and are well known to be relevant for the study aims. They should be considered when performing the P-S weighting.

Furthermore, the following minor revisions have to be checked:

1) Figures 2 and 3 have two identical labels, unlike figure 1 that has a short title and a more detailed label.

2) In figure 1 label the sentence “Step 1” isn’t complete and the following sentence is entitled “Table 2” (I guess instead of “Step 2”). Please check.

3) The Methods section is too wordy, please be more brief.

4) Table 6 is too wordy and very difficult to be read. Tables should summarize data. I suggest to report the primary outcomes and results only of each cited study and to discuss any contradictions or peculiar findings in the Discussion section.

Author Response

Comments for Reviewer 1 :

Dear reviewer #1, Thank you for reviewing our article and for your constructive comments. Please find our answers to your questions, comments and suggestions. We hope that this answers your questions, remarks and suggestions.

In the present paper Authors reported on a single center large series of NSCLC patients who underwent minimally invasive Segmentectomy or lobectomy. They retrospectively compared the outcomes of VATS vs RATS approaches, in terms of disease free survival, overall survival and time to recurrence. No significant difference was observed in each study endpoint. 

The study population seems to be well balanced and numbers are acceptable to pursue the study aims. However, I have a few concerns about this study methodology:

1) Although the 844 reported surgical cases were performed by seven surgeons, the majority of RATS procedures (97.8%) were performed by two of them only. Similarly, four surgeons performed 96.1% of VATS. Were the first two surgeon included in the latter four? If not, the present paper appears to be a comparison between surgeon and their personal techniques rather than between two different kind of surgical approach.

Thank you for that comment. The two surgeons are included in the latter four. The two first surgeons performed both VATS and RATS but the VATS/RATS ratio was unbalanced (more RATS for surgeon #1 and more VATS for surgeon #2). The following table shows the distribution of interventions among all surgeons.

VATS

RATS

Surgeon #1

144 (29.9%)

290 (80.1%)

Surgeon #2

156 (32.4%)

64 (17.7%)

Surgeon #3

110 (22.8%)

1 (0.3%)

Surgeon #4

53 (11%)

6 (1.7%)

Surgeon #5

10 (2.1%)

1 (0.3%)

Surgeon #6

6 (1.2%)

0 (0%)

Surgeon #7

3 (0.6%)

0 (0%)

Total

482 (100%)

362 (100%)

In order to take in account, the surgeon effect, we added the surgeon variable to the propensity score in the revised analysis. This makes sure that the technique, rather than the surgeon, is compared. However, it should reduce the statistical power, because surgeons who performed mainly one technique (surgeons #3 to #7) will be practically excluded.

We also described the activity of the most active surgeons to clarify that point, in the manuscript:

“Two surgeons (#1: n=290; #2: n=64) performed 354 (97.8%) RATS procedures. Four surgeons (#1: n=144; #2: n=156, #3: n=110; #4: n=53) performed 463 (96.1%) VATS procedures.

2) Page 4, line 160: TNM staging were converted to the 7th edition. I can’t understand why. Please convert to the most recent one (the 8th).

Thank you for that comment. Our pathologists used the 7th edition until july 2017. Since then, they used the 8th edition. The 8thedition is more fine grained than the 7th edition, making the retrospective conversion from the 7th edition to the 8th edition very hard, with a risk of measurement bias. From the 8th edition to the 7th edition, conversion could be performed automatically in most cases with less measurement bias. In a few cases, the conversion could not be automatic. For instance a pT3 in the 8th edition may be a pT2b or pT3 depending on the anatomical invasion (e.g. chest wall) and presence or absence of satellites nodules in the same lobe. With the help of the pathologist report conclusion, the conversion could be performed in almost all cases. In the last few cases, it was imputed by the most probable value.

3) Page 5, lines 173-183: here the covariates used for propensity-score weighting are reported. Some important variables such as smoking history and histology are lacking. Both of them are unbalanced between study groups and are well known to be relevant for the study aims. They should be considered when performing the P-S weighting.

Thank you. We added three variables to the propensity score: the smoking status, histology and surgeon.

We had not perform a heavy data management on the smoking status because it was a secondary variable. Since it is now included in the propensity score, we performed more data management and found that the four patients with missing smoking status were all smokers, as the “tobacco dependency” was listed in their comorbidities. Otherwise, there were no inconsistency between “tobacco dependency” comorbidities and smoking status. The table has been updated accordingly.

Furthermore, the following minor revisions have to be checked:

1) Figures 2 and 3 have two identical labels, unlike figure 1 that has a short title and a more detailed label.

Labels and titles were redundant. We completely deleted the on-figure labels. Moreover, we improved the adherence to the Instructions for Authors of Cancers, by using short explanatory titles.

2) In figure 1 label the sentence “Step 1” isn’t complete and the following sentence is entitled “Table 2” (I guess instead of “Step 2”). Please check.

To follow the Instructions for Authors, we removed the long description of figure 1. It is shortened to: “Multimodal approach to minimally invasive personalized tailored segmentectomy in 3 steps”. The figure is aimed at being self-explanatory.

3) The Methods section is too wordy, please be more brief.

We shortened this section by 25% approximatively.

4) Table 6 is too wordy and very difficult to be read. Tables should summarize data. I suggest to report the primary outcomes and results only of each cited study and to discuss any contradictions or peculiar findings in the Discussion section. 

Thank you for that comment. We removed all short-term outcomes from this table but we keeped hazard ratios of disease-free survival and overall survival. We also simplified and standardized the description of study settings.

Lymph node upstaging is one of the main factor that could affect long-term survival and also reported in articles in the first version of the table 6 and this point is already discussed in the discussion. Contradictions were discussed in the first version of the manuscript.

**Additional modifications**

As we reanalyzed the data, we noticed some minor errors in our analyses, without significant impact on results. We fixed these errors in the new manuscript:

  • The age had been omitted from the Charlson Comorbidity Index calculation, leading to an underestimation
  • The quality of long-term follow-up data was guaranteed up to the December 31st, 2019 but not beyond this date. All data beyond this date is properly censored in the new manuscript.
  • There were a few missing data (n=4) on the smoking status. This could be retrieved from the comorbidities (labeled as “tobacco dependency”).
  • A few errors in the histological types that had been fixed during the database review were improperly ignored in the previous manuscript. They are fixed in the new manuscript.
  • The paragraph “Per-operative and short-term post-operative outcomes” contained a small error of interpretation of a result of Table 3: the difference was about the distribution of all conversions to thoracotomy but had been interpreted as related only to conversions for operative complications.

With multiple manuscript editing and database cleanups, human error occur. Constant vigilance is required.

Reviewer 2 Report

This manuscript entitled Long-term survival following minimally invasive lung cancer surgery: comparing robotic-assisted and video-assisted surgery aimed to assess whether RATS increased disease-free survival compared with VATS for lobectomy and segmentectomy in early-stage non-small cell lung cancer. The results showed  the adjusted 5-year DFS for lobectomy and the adjusted 3-year DFS for segmentectomy were not statistically different in these two surgical approach. The report concludes that RATS failed to show its superiority over VATS for resectable NSCLC.

This manuscript is well written with high significance of content and high scientific soundness.

Regarding the selection for surgical procedure in multidisciplinary meeting in Inclusion criteria (Line 80), suggest the authors to describe the decision-making mechanisms clearly and discuss the possible impacts on the results in Discussion. 

Author Response

Comments for Reviewer 2 :

Dear reviewer #2, Thank you for reviewing our article and for your constructive comment. Please find our answers to your comment and suggestion. We hope that this answers your comment and suggestion.

This manuscript entitled Long-term survival following minimally invasive lung cancer surgery: comparing robotic-assisted and video-assisted surgery aimed to assess whether RATS increased disease-free survival compared with VATS for lobectomy and segmentectomy in early-stage non-small cell lung cancer. The results showed  the adjusted 5-year DFS for lobectomy and the adjusted 3-year DFS for segmentectomy were not statistically different in these two surgical approach. The report concludes that RATS failed to show its superiority over VATS for resectable NSCLC.

This manuscript is well written with high significance of content and high scientific soundness.

Regarding the selection for surgical procedure in multidisciplinary meeting in Inclusion criteria (Line 80), suggest the authors to describe the decision-making mechanisms clearly and discuss the possible impacts on the results in Discussion. 

Thank you for that comment.

In France, all cases having surgically operable or inoperable cancers are discussed in multidisciplinary meetings in order to guarantee that the medical and surgical choices are performed in accordance to national and international guidelines. This is a legal obligation. The chemotherapy, radiotherapy and surgery options are discussed but not the specific surgical approach, as long as the approach guarantees a macroscopically complete resection and a lymph node dissection conforming to guidelines with conversion to thoracotomy if needed. The RATS or VATS approach is mainly dependent on the surgeon habits and availability of the da Vinci platform. Indeed, this robotic platform was available only 6 days a month during all the study period. Therefore, there is no risk of confounding by indication due to the multidisciplinary meetings. The indication bias is mainly due to the surgeon, with an overall preference for RATS for segmentectomy and VATS for lobectomy. There was also a major surgeon effect, with surgeon #1 performing more RATS than VATS (n=290 vs n=144) and surgeon #2 performing fewer RATS than VATS (n=64 vs n=156). These two main indication bias are controlled by two means. First, segmentectomy and lobectomy are analyzed separately. Second, the new statistical analysis adjust on the surgeon. As the study is observational, there may be some residual confounding bias.

In order to clarify those facts but to keep the manuscript concise, we added this paragraph to the methods section:

“The cancer treatment (radiotherapy, chemotherapy, surgery) was chosen in multidisciplinary meetings in accordance with guidelines. The choice between VATS and RATS was at the discretion of the surgeon and depended on surgeon's preference and availability of the robot: 6 days a month on the study period.”

And changed the paragraph about confounding bias in the Discussion:

“As this study is observational, confounding by indication was possible. The indication of VATS or RATS mainly depended on surgeon’s preferences and the type of surgery, with a preference for RATS for segmentectomy and for VATS for lobectomy. Since segmentectomy and lobectomy were analyzed separately and the surgeon was included in the propensity score, these main indication biases were canceled. However, we noticed a significant difference of ECOG performance status between RATS and VATS in the lobectomy group suggesting that other confounders were possible. There were adjustments on main prognostic factors, including the ECOG performance status, but a residual confounding bias is possible.”

**Additional modifications**

As we reanalyzed the data, we noticed some minor errors in our analyses, without significant impact on results. We fixed these errors in the new manuscript:

  • The age had been omitted from the Charlson Comorbidity Index calculation, leading to an underestimation
  • The quality of long-term follow-up data was guaranteed up to the December 31st, 2019 but not beyond this date. All data beyond this date is properly censored in the new manuscript.
  • There were a few missing data (n=4) on the smoking status. This could be retrieved from the comorbidities (labeled as “tobacco dependency”).
  • A few errors in the histological types that had been fixed during the database review were improperly ignored in the previous manuscript. They are fixed in the new manuscript.
  • The paragraph “Per-operative and short-term post-operative outcomes” contained a small error of interpretation of a result of Table 3: the difference was about the distribution of all conversions to thoracotomy but had been interpreted as related only to conversions for operative complications.

With multiple manuscript editing and database cleanups, human error occur. Constant vigilance is required.

Reviewer 3 Report

Excellent paper in a field very actual. Is RATS surgery more effective than VATS surgery?

The research is well-conducted and the result give an answer to the question.

The only bias is that the study is retrospective. A prospective study is well-come but it requires longer time.

Author Response

Comments for Reviewer 3 :

Dear reviewer #3, Thank you for reviewing our article and for your constructive comment. We hope that this answers your suggestion.

Excellent paper in a field very actual. Is RATS surgery more effective than VATS surgery? The research is well-conducted and the result give an answer to the question. The only bias is that the study is retrospective. A prospective study is well-come but it requires longer time.

Thank you for that positive comment. A large multicenter randomized study would be ideal, but as you said, it would require more time. Moreover, feasibility is not guaranteed: it would require a high inclusion rate on a very long time period with motivated surgeons trained to both techniques.

**Additional modifications**

As we reanalyzed the data, we noticed some minor errors in our analyses, without significant impact on results. We fixed these errors in the new manuscript:

  • The age had been omitted from the Charlson Comorbidity Index calculation, leading to an underestimation
  • The quality of long-term follow-up data was guaranteed up to the December 31st, 2019 but not beyond this date. All data beyond this date is properly censored in the new manuscript.
  • There were a few missing data (n=4) on the smoking status. This could be retrieved from the comorbidities (labeled as “tobacco dependency”).
  • A few errors in the histological types that had been fixed during the database review were improperly ignored in the previous manuscript. They are fixed in the new manuscript.
  • The paragraph “Per-operative and short-term post-operative outcomes” contained a small error of interpretation of a result of Table 3: the difference was about the distribution of all conversions to thoracotomy but had been interpreted as related only to conversions for operative complications.

With multiple manuscript editing and database cleanups, human error occur. Constant vigilance is required.

Round 2

Reviewer 1 Report

I would have appreciated to see the TNM converted to the 8th edition. However, Authors have done a proper and extensive revision of the paper according to my comments. Hoping to read from you more, many thanks.